# Achieving Good Temperature Stability of Dielectric Constant by Constructing Composition Gradient in (Pb_1−x_,La_x_)(Zr_0.65_,Ti_0.35_)O_3_ Multilayer Thin Films

**DOI:** 10.3390/ma15124123

**Published:** 2022-06-10

**Authors:** Ming Wu, Yanan Xiao, Yu Yan, Yongbin Liu, Huaqiang Li, Jinghui Gao, Lisheng Zhong, Xiaojie Lou

**Affiliations:** 1State Key Laboratory of Electrical Insulation and Power Equipment, Xi’an Jiaotong University, Xi’an 710049, China; wuming@xjtu.edu.cn (M.W.); xiaoyanan@stu.xjtu.edu.cn (Y.X.); yanyu13@stu.xjtu.edu.cn (Y.Y.); liuyongbin@xjtu.edu.cn (Y.L.); lhqxjtu@xjtu.edu.cn (H.L.); 2Frontier Institute of Science and Technology, and State Key Laboratory for Mechanical Behavior, Xi’an Jiaotong University, Xi’an 710049, China; xlou03@xjtu.edu.cn

**Keywords:** PLZT, dielectric constant, temperature stability, ferroelectric thin film

## Abstract

Ferroelectrics with a high dielectric constant are ideal materials for the fabrication of miniaturized and integrated electronic devices. However, the dielectric constant of ferroelectrics varies significantly with the change of temperature, which is detrimental to the working stability of electronic devices. This work demonstrates a new strategy to design a ferroelectric dielectric with a high temperature stability, that is, the design of a multilayer relaxor ferroelectric thin film with a composition gradient. As a result, the fabricated up-graded (Pb,La)(Zr_0.65_,Ti_0.35_)O_3_ multilayer thin film showed a superior temperature stability of the dielectric constant, with variation less than 7% in the temperature range from 30 °C to 200 °C, and more importantly, the variation was less than 2.5% in the temperature range from 75 °C to 200 °C. This work not only develops a dielectric material with superior temperature stability, but also demonstrates a promising method to enhance the temperature stability of ferroelectrics.

## 1. Introduction

A dielectric capacitor is one of the basic units for assembling electronic devices, and its temperature stability determines the operating temperature range of the electronic devices [1,2]. In some cases, the upper limitation of the operating temperature is firstly considered, such as the application in the downhole oil and gas industry, high-power electronics and hybrid electric vehicle (HEV) [3,4]. For instance, in the power inverters of HEV, the capacitors must work at around 140 °C [5]. The capacitors used in the downhole industry are operated at temperature above 150 °C [6]. High-frequency SiC MOSFET can heat up to 200 °C even under normal load, so nearby electronic capacitors must be able to withstand this temperature [7]. In the EIA standard for class II dielectrics for capacitor applications (EIA-198-1-F-2002), 200 °C is the high temperature limit to evaluate the thermal stability of high-temperature dielectric materials. Therefore, it is exigent to develop dielectric materials with good temperature stability to satisfy the increasing demand from industry applications.

Polar materials such as ferroelectrics possess a large relative dielectric constant (*ε_r_*), which is beneficial to the miniaturization and integration of dielectric capacitors [8]. However, the dielectric constant of ferroelectrics is very sensitive to the change of temperature, especially around the Curie temperature where the ferroelectric-to-paraelectric phase transition occurs [9]. Inducing a relaxor state by chemical doping has been proved to be an effective method to enhance the temperature stability of the dielectric constant in ferroelectrics [10,11]. In a relaxor ferroelectric, the long-range order of the polarization in the ferroelectric is disturbed by the doped elements and induces polar nanoregions, which contribute to a diffused phase transition and therefore a flat dielectric peak over a wide temperature range [12,13]. For instance, by using elements doping, Ren et al. introduced a relaxor state in the (Na_0.5_,Bi_0.5_)TiO_3_-based ceramic. As a result, in the temperature range from −90 to 320 °C, the variation of permittivity related to the permittivity at 25 °C was less than ±15% [14]. Another strategy to improve the dielectric temperature stability is to design multilayer structures by using ferroelectrics with successive Curie temperatures [15]. Recently, Gao et al. designed a laminated structure of tricritical ferroelectrics with successive Curie temperatures and achieved superior dielectric stability in the temperature range from 30 °C to 85 °C [16].

In this work, a combined strategy is adopted to enhance the dielectric temperature stability of ferroelectric thin film capacitors, that is, we design a multilayer ferroelectric thin film capacitor by using a relaxor ferroelectric with different Curie temperatures. The typical relaxor ferroelectrics (Pb_1−x_,La_x_)(Zr_0.65_,Ti_0.35_)O_3_ (abbreviated as PLZT100X, X = 0.06, 0.08, 0.10 and 0.12) were selected as the starting composition. The Curie temperature of the (Pb_1−x_,La_x_)(Zr_0.65_,Ti_0.35_)O_3_ decreases almost linearly with the increase of the La doping concentration, from about 500 K when X = 0.06 to about 300 K when X = 0.12 [17]. The schematics of the multilayer structure are shown in Figure 1. In the case of a La concentration gradually increasing along the growth direction (Figure 1a), the structure is called up-graded. Otherwise, it is called down-graded (Figure 1b). The up-graded and down-graded multilayer ferroelectric thin films are deposited by sol–gel method and the thermal stability of its dielectric properties were studied in the temperature range from 30 to 200 °C. The results show that the up-graded structure has superior thermal stability compared with that of the down-graded structure and other thin films without a composition gradient, which is very suitable for its application in electronic devices with the requirement of withstanding high operating temperature.

## 2. Experimental Procedure

The ferroelectric thin films were deposited by the sol–gel method and the overall process is shown in Figure 1c. The precursor solutions with compositions of (Pb_1−x_,La_x_)(Zr_0.65_,Ti_0.35_)O_3_ (X = 6, 8, 10, 12) were synthesized using lead acetate trihydrate (Pb(CH_3_COO)_2_·3H_2_O), lanthanum acetate (La(CH_3_COO)_3_), zirconium-n-propoxide (Zr(OC_3_H_7_)_4_) and titanium isopropylate (Ti(OCH(CH_3_)_2_)_4_) as the raw materials. Acetic acid and 2-methoxyethanol were mixed with the ratio of 1:4 and stirred for 60 min as cosolvent. A 10% excess of lead acetate trihydrate was used to compensate the lead loss and to prevent the formation of pyrochlore phase in the films during crystallization. The final concentration of the solution was 0.4 M. After aging for 24 h, the PLZT thin films were deposited on the Pt/Ti/SiO_2_/Si substrate by a multiple-step spin-coating process. Each film was spin coated at a speed of 3500 rpm for 30 s by using a spin coater. Then, the wet film was baked at 350 °C for 2 min to remove the solvent and at 500 °C for 2 min to decompose the organic matter, and subsequently annealed at 650 °C for the formation of the perovskite phase. For the up-graded PLZT films, the spin coating and heat treatment were repeated four times with the La content in the precursor solutions varying from 6 mol % at the substrate end to 12 mol % at the top surface. Films with opposite gradients were called down-graded films. Platinum top electrodes (90 μm × 90 μm) were sputtered through a copper mask for electrical measurement.

The phase structure of the PLZT thin films were analyzed by an X-ray diffractometer (XRD, D8 Advance, Bruker, MA, USA) with a step of 0.02° from 20° to 60°. The surface microstructure and cross section of the RFE thin films were determined by a scanning electron microscope (SEM ZEISS GeminiSEM500, Oberkochen, Germany). The frequency and temperature dependence of the dielectric permittivity and dielectric loss were measured using an LCR meter (Agilent E4980A, CA, USA) associated with a temperature controller (Linkam THMS600, Tadworth, UK).

## 3. Results and Discussion

The crystal structure of the multilayer PLZT thin films were studied by XRD and the results are shown in Figure 2. It can be found that, for both up-graded and down-graded PLZT thin films, all the diffraction peaks (except the peaks from the Pt/Ti/SiO_2_/Si substrate) could be indexed within a perovskite structure, which suggested a pure phase for both structures. The grain size of the PLZT thin films were calculated by the Scherrer equation based on the X-ray diffraction patterns [18]. It was found that the average grain size of the up-graded thin film was around 50.6 nm, which was smaller than that of the down-graded thin film (which was around 65.9 nm). Both up-graded and down-graded PLZT thin films showed several broadened diffraction peaks, which indicated a polycrystal characteristic. In addition, the preferred crystallographic direction for both samples was (111), as (111) was the strongest diffraction peak. In a word, the crystal properties of the PLZT thin films were not affected directly by the composition gradient.

The surface and cross section morphologies of the PLZT thin films are shown in Figure 3. It can be seen that both up-graded and down-graded thin films show a very smooth and dense microstructure, without notable pores and cracks, which demonstrates a high crystalline quality for both samples. The thickness of the up-graded and down-graded thin films is about 390 nm and 410 nm, respectively.

The relative dielectric constant (*ε_r_*) and dielectric loss (*tan**δ*) for the up-graded and down-graded PLZT thin films were measured at the frequencies of 500 Hz, 1 kHz, 10 kHz, 100 kHz and 1 MHz in the temperature range from 30 °C to 200 °C, and the results are presented in Figure 4a,b. We found that the dielectric constant of the up-graded thin film (around 700 in the measuring temperature range) was slightly smaller than that of the down-graded thin film (around 750 in the measuring temperature range), which was due to the relatively smaller grain size of the up-graded thin film compared with that of the down-graded thin film, as observed in Figure 3. According to F. C. Kartawidjaja et al. [19], the larger grain size always possesses a large dielectric constant due to the enhanced movement of the domain wall, which is also consistent with the observed results in the PLZT thin film. In addition, the dielectric constant of the PLZT thin film was much smaller than that of PLZT ceramics with a similar composition (the dielectric constant of the PLZT9/65/35 ceramic is around 3400) [20], which was due to the much smaller grain size of the PLZT thin film and the existence of an interface stress and interface dead layer between the PLZT thin film and the deposition substrate [21]. For the up-graded PLZT thin films, as shown in Figure 4a, the relative dielectric constant decreased with the increase of frequency and showed a broad peak in the measuring temperature range. In addition, the temperature of the peak increased with the increase of the measuring frequency, which indicated the diffuse phase transition of the typical relaxor ferroelectric. The very broad and flat dielectric peak revealed a very good thermal stability of the up-graded PLZT thin films. The dielectric loss kept below 0.01 at lower frequencies and in the lower temperature region. When the frequency increased to 1 MHz, the dielectric loss significantly increased to around 0.06, which was due to the delayed reorientation of the ferroelectric polarization at an elevated switching frequency. For the dielectric loss measured below 1 MHz, it remarkably increased when the temperature went beyond 150 °C, and the lower the measuring frequency, the faster the dielectric loss increased. This feature arises from the increased carrier’s mobility with the increased temperature, which leads to larger conductance losses. The down-graded PLZT thin film showed a similar tendency as the up-graded one but possessed a slightly increased relative dielectric constant with the sacrifice of thermal stability to some degree.

To further quantitatively evaluate the temperature stability of the dielectric constant of the up-graded and down-graded PLZT thin films, the variation of the dielectric constant at different temperatures related to the dielectric constant at 30 °C was calculated as follow [22]:(1)$$\begin{array}{c} \;\frac{∆\varepsilon_{r}}{\varepsilon_{30\;{}^{\circ}C}}=\frac{\varepsilon_{r}-\varepsilon_{30\;{}^{\circ}C}}{\varepsilon_{30\;{}^{\circ}C}}\times 100\% \end{array}$$
where *ε*_30_
_°__C_ is the dielectric constant at 30 °C, *ε_r_* is the dielectric constant at different temperatures and ∆*ε_r_* is the variation of the dielectric constant related to 30 °C. The results are shown in Figure 4c,d. In order to better illustrate the effect of the composition gradient design on the up-graded and down-graded PLZT thin films, the temperature stability of the dielectric constant of the PLZT6, PLZT8, PLZT10 and PLZT12 thin films were also calculated and compared together in Figure 4c,d. The dielectric constant of PLZT6, PLZT8, PLZT10 and PLZT12 are copied from previously published data [23].

Figure 4c shows the temperature stability of the PLZT thin films at 1 kHz. It can be seen that the variation of the dielectric constant for all the PLZT thin films are less than 20% in the measuring temperature range from 30 °C to 200 °C, which is due to the relaxor characteristic. Notably, the up-graded PLZT thin film has the best temperature stability, with a variation of less than 7% in the measuring temperature range. More importantly, the up-graded PLZT thin film shows superior temperature stability in the temperature range from 75 to 200 °C, with a variation of the dielectric constant of less than 2.5%. The temperature stability of the dielectric constant of the PLZT thin films were also compared at 10 kHz (Figure 4d), which gave a very similar result to that at 1 kHz. These results suggest that, by using a composition gradient design to construct the relaxor ferroelectric multilayer thin film, we can further improve the temperature stability of the relaxor ferroelectric materials, which is very beneficial for its application in high-temperature electronic devices.

The temperature stability of the dielectric constant of the proposed up-graded PLZT thin film was also compared with other reported materials, as shown in Figure 5. The materials are roughly divided into three classes. The first class consists of lead zirconate titanate (PZT)-based materials, including typical relaxor ferroelectrics (Pb_0.92_La_0.08_)(Zr_0.65_Ti_0.35_)O_3_ (PLZT8/65/35) [24], (Pb_0.89_La_0.11_)(Zr_0.70_ Ti_0.30_)O_3_ (PLZT11/70/30) [25] and a ferroelectric thin film with a Pb(Zr_0.4_Ti_0.6_)O_3_/BaZr_0.2_Ti_0.8_O_3_/Pb(Zr_0.4_Ti_0.6_)O_3_ (PZT/BT/PZT) [26] multilayer structure. The second class consists of barium titanate (BT)-based materials, including three compounds, that is, 0.7BaTiO_3_-0.3Bi(Mg_0.5_Zr_0.5_)O_3_ (70BT-30BMZ) [27], (1−x)BaTiO_3_-xBi(Zn_0.5_Y_0.5_)O_2.75_ (BT-BZY) [28] and BaTiO_3_-(Bi_0.5_Na_0.5_)TiO_3_-Bi(Mg_0.5_Zr_0.5_)O_3_-Ba(Fe_0.5_Nb_0.5_)O_3_ (BT-BNT-BMZ-BFN) [29]. The third class consists of sodium bismuth titanate (BNT)-based materials, including (1−x) [0.94(0.75Bi_0.5_Na_0.5_TiO_3_-0.25NaNbO_3_)-0.06BaTiO_3_]-xCaZrO_3_ (BNT-NN-BT-CZ) [30], (1−x)[0.90Na_0.5_Bi_0.5_TiO_3_-0.10BiAlO_3_]-xNaNbO_3_ (BNT-BA-NN) [14] and (1−x)(0.94Bi_0.5_Na_0.5_ TiO_3_-0.06BaTiO_3_)-xK_0.5_Na_0.5_NbO_3_ (BNT-BT-KNN) [31]. The temperature stability of the dielectric constant of the above materials are revealed by the change of color in Figure 5, with green representing a very good temperature stability while pale yellow and dark blue representing positive and negative variations of the dielectric constant related to that measured at 30 °C. It is seen that compared with other materials, the up-graded PLZT thin films show an almost unchanged green color in the temperature range from 30 to 200 °C, which indicates a superior temperature stability in this temperature range.

## 4. Conclusions

In this work, relaxor ferroelectric PLZT thin films with a composition gradient were deposited by the sol-gel method and the temperature stability of the dielectric properties were studied. It was found that the up-graded PLZT had a superior temperature stability of the dielectric constant compared with a pure PLZT composition, as well as other reported dielectric materials. The variation of the dielectric constant of the up-graded PLZT thin film was less than 7% in the temperature range from 30 °C to 200 °C, and less than 2.5% in the temperature range from 75 °C to 200 °C. This superior performance was attributed to the composition gradient in the up-graded multilayer PLZT thin film. This work not only developed a dielectric material with superior temperature stability, but more importantly, demonstrated the feasibility to develop thermal stable dielectric materials using compositional gradient in relaxor ferroelectrics.

## Figures and Tables

**Figure 1 materials-15-04123-f001:**
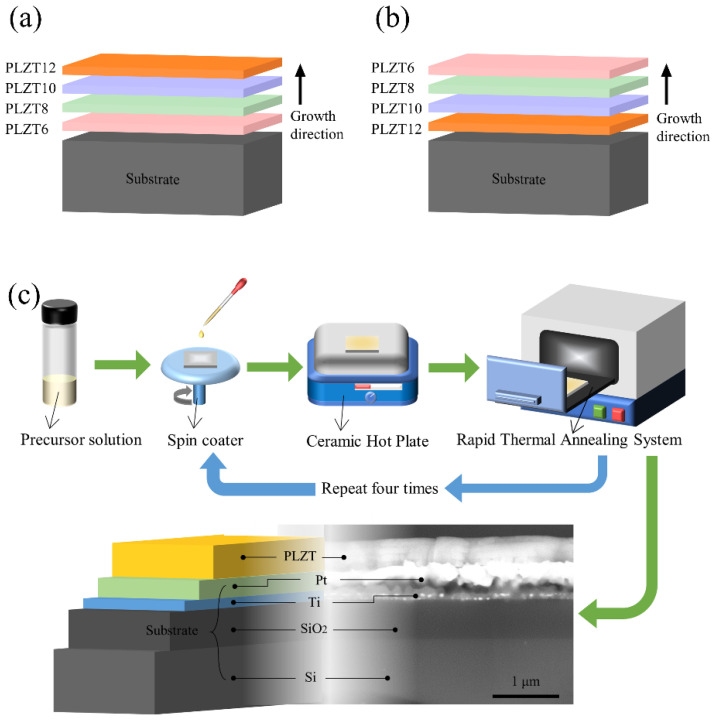
Schematic of the (**a**) up-graded and (**b**) down-graded multilayer ferroelectric thin film. (**c**) Process of fabricating the ferroelectric thin films.

**Figure 2 materials-15-04123-f002:**
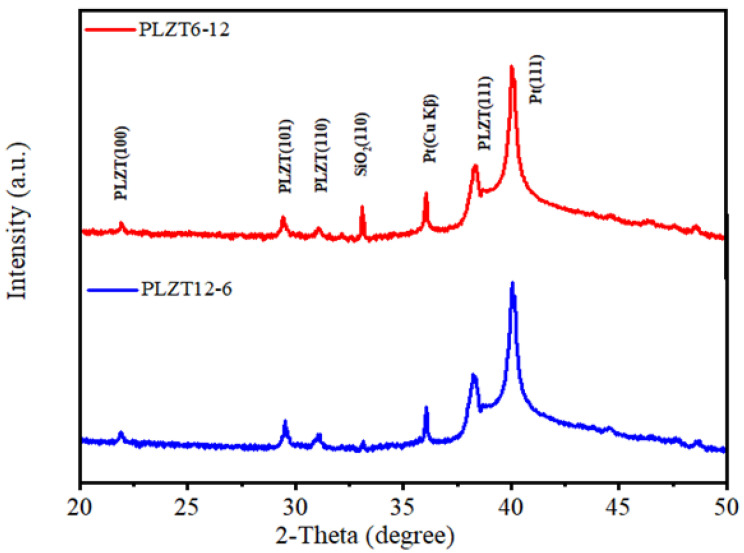
XRD patterns of the up-graded and down-graded PLZT thin films.

**Figure 3 materials-15-04123-f003:**
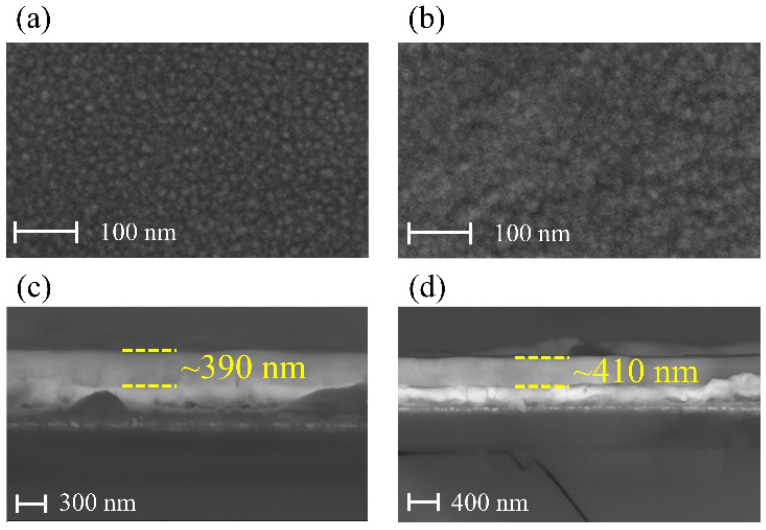
Surface morphologies for (**a**) up-graded and (**b**) down-graded PLZT thin films. Cross section morphologies for (**c**) up-graded and (**d**) down-graded PLZT thin films.

**Figure 4 materials-15-04123-f004:**
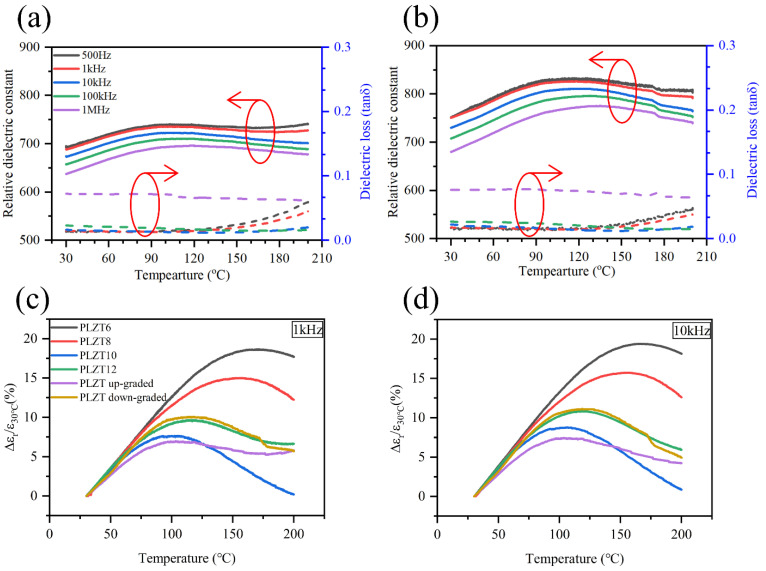
Temperature dependent dielectric constant of the (**a**) up-graded and (**b**) down-graded PLZT thin films. (**c**) Temperature stability of the PLZT thin film when measured at (**c**) 1 kHz and (**d**) 10 kHz.

**Figure 5 materials-15-04123-f005:**
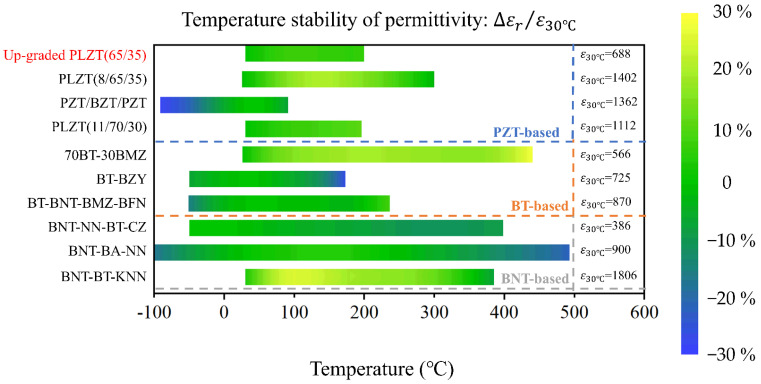
Comparison of temperature stability of dielectric constant between up-graded PLZT thin film and other dielectric materials.

## Data Availability

The data used to support the findings of this study are available from the corresponding author upon reasonable request.

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
