# Peer review of "Achieving Good Temperature Stability of Dielectric Constant by Constructing Composition Gradient in (Pb1−x,Lax)(Zr0.65,Ti0.35)O3 Multilayer Thin Films"

_materials, 2022, doi:10.3390/ma15124123_

Round 1

Reviewer 1 Report

 Dear editor,

The manuscript  “Achieving good temperature stability of dielectric constant by constructing composition gradient in (Pb1-x,Lax)(Zr0.65,Ti0.35)O3 multilayer thin films’ with Manuscript ID: materials-1740342, by Ming Wu et al. was evaluated.

 This work demonstrates a new strategy to design ferroelectric dielectric with high temperature stability, that is, design multilayer relax or ferroelectric thin film with composition gradient. As a result, the fabricated up-graded (Pb,La)(Zr0.65Ti0.35)O3 multilayer thin film shows superior temperature stability of dielectric constant, with variation less than 7% in the temperature range from 30 °C to 200 °C.

 I recommend major revisions as:

  1. In the results and discussion (part. 3). authors should clearly discuss the structure pattern of the related materials, also, they should explain the relation between the dielectric results and microstructure and structure parameters of the up-graded and down-graded PLZT thin films.

This can be addressed clearly.

  1. Authors report in the Figure 4. “Temperature dependent dielectric constant of the (a) up-graded and (b) down graded PLZT thin films. (c) Temperature stability of the PLZT thin film when measured at (c) 1 kHz and (d) 10 kHz”

 I don’t see any discussion about the relationship between Temperature and dielectric constant.! Is there any equation? This can be also evaluated and discussed in the manuscript.

  1. about the choice of measuring temperature range from 30 °C to 200 °C, Is there any reason? It should be clearly stated.

  1. also, in my view, authors should clearly define the dependence of the dielectric constant (εr) and dielectric loss (tanδ) to the frequency and discuss them.

Reviewer 2 Report

Please see the enclosed file.

Reviewer 3 Report

Please find the review details in the attached file.

Round 2

Reviewer 1 Report

  1. The authors tried to modify the manuscript as per my request. However, I couldn’t see any brilliant modification and discussion about the relation between the dielectric results and microstructure parameters (crystallite size , lattice parameters …) of the up-graded and down-graded PLZT thin films, in the results and discussion (part. 3). For example how they deduced the grain size? !!

I think this can be enlarged and discussed in a suitable way.
